# Effect of hydrophobic cations on the oxygen reduction reaction on single–crystal platinum electrodes

Tomoaki Kumeda[1], Hiroo Tajiri[2], Osami Sakata [3], Nagahiro Hoshi[1] & Masashi Nakamura [1]

Highly active catalysts for the oxygen reduction reaction are essential for the widespread and economically viable use of polymer electrolyte fuel cells. Here we report the oxygen reduction reaction activities of single–crystal platinum electrodes in acidic solutions containing tetra-alkylammonium cations with different alkyl chain lengths. The high hydrophobicity of a tet-raalkylammonium cation with a longer alkyl chain enhances the oxygen reduction reaction activity. The activity on Pt(111) in the presence of tetra–n–hexylammonium cation is eight times as high as that without this cation, which is comparable to the activities on $Pt_3Co(111)$ and $Pt_3Ni(111)$ electrodes. Hydrophobic cations and their hydration shells destabilize the adsorbed hydroxide and adsorbed water. The hydrophobic characteristics of non–specifically adsorbed cations can prevent the adsorption of poisoning species on the platinum electrode and form a highly efficient interface for the oxygen reduction reaction.

[1] Department of Applied Chemistry and Biotechnology, Graduate School of Engineering, Chiba University, Yayoi-cho 1-33, Inage-ku, Chiba 263-8522, Japan. [2] Research and Utilization Division, Japan Synchrotron Radiation Research Institute/SPring–8, Kouto 1-1-1, Sayo-gun, Hyogo 679-5148, Japan. [3] Synchrotron X-ray Group and Synchrotron X-ray station at Spring-8, National Institute for Materials Science, Kouto 1-1-1, Sayo-gun, Hyogo 679-5148, Japan. Correspondence and requests for materials should be addressed to M.N. (email: mnakamura@faculty.chiba-u.jp)

The development of highly active catalysts for the oxygen reduction reaction (ORR) is a prominent issue in terms of decreasing Pt loading in cathode catalysts for polymer electrolyte fuel cells (PEFCs). Model electrocatalysts with atomically well-defined surfaces have provided detailed information regarding activation sites, optimal atomic arrangement, and the surface composition of catalysts, which have been applied to nanomaterials for practical use. In the case of Pt-based materials, the introduction of heterometals drastically improves ORR activity, depending on the surface structure[1–3]. PtM (M = Ni, Co, etc.) bimetal alloys perturb the density of state (DOS) of the d-band and compress the lattice strain of the surface layer[4,5]. Consequently, the ORR activities on PtM(111) are increased by one order of magnitude as high as that of Pt(111) electrode[6,7]. In many heterogeneous catalysts, the catalytic activity has a volcano-shaped relationship with the atomic arrangement and alloy composition of the substrate. The variation of the d-band DOS of the substrate modifies the adsorption energy of the reaction products and the intermediate species. Therefore, the center position of the d-band DOS strongly correlates with the ORR activity. $Pt_3Ni$ and $Pt_3Co$ are located near the top in the volcano-shaped dependency between the activity and the d-band center[4]. The introduction of step and kink structures also activates the ORR; the high-index planes of Pt give the maximum activity for the ORR on the surfaces with 3−4 atomic rows of (111) terrace[8–10]. The catalyst development through substrate modification has reached a maturity stage, and other approaches are required for further enhancement of the ORR activity.

The activity of electrochemical reactions is affected by the solvent and electrolyte ions as well as the substrate. At the electrode/electrolyte interface, such solution species construct the electric double layer (EDL), which governs the ORR activity on Pt surfaces significantly. It is well known that strongly adsorbed anions, such as halide and sulfate ions, on the Pt surface inhibit the ORR severely[11–13]. The adsorbed hydroxide ($OH_{ad}$) species is formed on Pt surfaces in electrolytic solutions containing ions that weakly interact with Pt surfaces[13]. The stability of $OH_{ad}$ depends on the surface structure and on the electronic state of the substrate[4,14–16]. Since $OH_{ad}$ is also an inhibitor for the ORR, the control of $OH_{ad}$ is a key factor in activating the ORR.

In the EDL, some hydrated cations located at the outer Helmholtz plane (OHP) are stabilized as non-specifically adsorbed species through non-covalent interactions, such as hydrogen bonds and electrostatic interactions[17]. Non-covalent interactions in the EDL shift the adsorption equilibrium and the phase transition potential of the adsorbed layer[18]. In alkaline media, the ORR activity of Pt(111) is improved exponentially as the hydration energy of the OHP cations decreases; the activity in CsOH is one order of magnitude higher than that in LiOH[19]. Cations with a high affinity toward oxygen species, such as $Li^+$, strongly stabilize $OH_{ad}$[20], which then deactivates the ORR. Therefore, appropriate control of the structure and hydrophobicity of the interfacial cations can improve ORR activity.

In PEFCs, which use proton-exchange membranes, the control of hydrophobicity is necessary in acidic solutions. In this study, we focus on tetraalkylammonium (TAA) cations of which the hydrophobicity and interfacial structure can be controlled by the alkyl chain length. Variation of the alkyl chain length can change the hydration structure of $TAA^+$ dramatically[21,22]. We have evaluated the ORR activity on single-crystal Pt electrodes in acidic solutions containing $TAA^+$ with different alkyl chains lengths. The interfacial structures have also been determined by in situ X-ray scattering and infrared (IR) spectroscopy measurements.

## Results

**Electrochemical characterization.** We used four types of $TAA^+$ with different alkyl chains lengths ($n$): tetramethylammonium ($TMA^+$) $n = 1$, tetraethylammonium ($TEA^+$) $n = 2$, tetra-$n$-butylammonium ($TBA^+$) $n = 4$, and tetra-$n$-hexylammonium ($THA^+$) $n = 6$. Figure 1a shows cyclic voltammograms (CVs) in 0.1 M $HClO_4$ containing $10^{-5}$ M $TBA^+$ and $10^{-6}$ M $THA^+$. The solubility of $TAA^+$ in acidic solution decreases with the increase of alkyl chain length. The concentrations of $TBA^+$ and $THA^+$ used in this study are approximately equal to the saturation. The CV data show the hydrogen adsorption/desorption region between 0.05 and 0.40 V, the double layer charging/discharging region between 0.40 V and 0.60 V, and the Pt oxidation region between 0.60 V and 0.90 V. In the presence of $TBA^+$ and $THA^+$, the onset potentials for the adsorption of hydrogen and Pt oxidation shift negatively and positively, respectively, whereas the CV data in solutions containing $10^{-3}$ M $TMA^+$ and $TEA^+$ are identical to those without $TAA^+$ (Supplementary Figure 1a). This indicates that $TAA^+$ with longer alkyl chains affects the adsorption of hydrogen and Pt oxidation, even at low concentration.

No IR absorption band for the alkyl chain of $TAA^+$ appears, as shown in Supplementary Figure 2. Previous IR study has revealed that the specific adsorption of $TAA^+$ on Pt does not occur above 0 V[23]. Therefore, the potential shift of hydrogen adsorption is not due to site blocking by specifically adsorbed $TAA^+$. The hydrogen adsorption step is governed by the proton-transfer process in the EDL, which depends on the conformation and hydrogen-bonding interactions of hydration water. Density functional theory (DFT) calculations suggest that hydrogen adsorption energy is affected by proton transfer in the water bilayer[24]. These results indicate that $TBA^+$ and $THA^+$ located near the electrode induce the reconstruction of interfacial water, and the inhibition of proton transfer in the EDL results in the potential shift for hydrogen adsorption. In the solutions containing $10^{-3}$ M $TMA^+$ and $TEA^+$, no potential shift appears even though the concentrations of $TMA^+$ and $TEA^+$ are larger by two or three orders of magnitude than those of $TBA^+$ and $THA^+$. Since the enthalpy of hydration water with $TAA^+$ is reduced by an increase of the alkyl chain length[25], the enhancement of hydrophobicity with longer alkyl chains strongly perturbs the interfacial water, changing the equilibrium potentials for the adsorption of hydrogen and Pt oxidation on Pt(111).

Figure 1b shows linear sweep ORR voltammograms in 0.1 M $HClO_4$ containing $TBA^+$ and $THA^+$ (voltammograms in $TMA^+$ and $TEA^+$ containing solutions are shown in Supplementary Figure. 1b). The specific activities ($j_k$) at 0.90 V in $TAA^+$-containing solutions are shown in Fig. 1c. The ORR activity increases in the sequence $THA^+ \gg TBA^+ > TEA^+ > TMA^+ \approx HClO_4$, and shows a correlation with the hydrophobicity of $TAA^+$. The ORR activity in $THA^+$ is eight times greater than that in 0.1 M $HClO_4$. The structural effects of the substrates on the ORR were investigated using the other typical index planes of Pt, as shown in Fig. 1d (CVs and linear sweep ORR voltammograms are shown in Supplementary Figure 3). There is no significant enhancement effect of $THA^+$ on Pt(100) and Pt(110). However, $THA^+$ enhances the activity on Pt(331) = 3(111)-(111), which gives the highest activity for the ORR on stepped surfaces, by 1.3 times. The specific activities on Pt(111) and Pt(331) in $THA^+$-containing solution are comparable with those of $Pt_3Co$ and $Pt_3Ni$[6,7].

At the initial stage of the Pt oxidation, the in-plane bending mode ($\delta_{PtOH}$) of $OH_{ad}$ is detected by IR spectroscopy. The band intensities of $\delta_{PtOH}$ at 0.90 V correlate to the ORR activities $j_k$ at 0.90 V on single-crystal Pt electrodes, confirming that $OH_{ad}$ inhibits the ORR[15,16]. The stability of the hydrogen-bonding network in the coadsorbed $OH_{ad} + H_2O$ layer affects the ORR activity on the (111) terrace, as described in the Discussion section. The approach of hydrophobic cation to the surface may causes the destabilization of

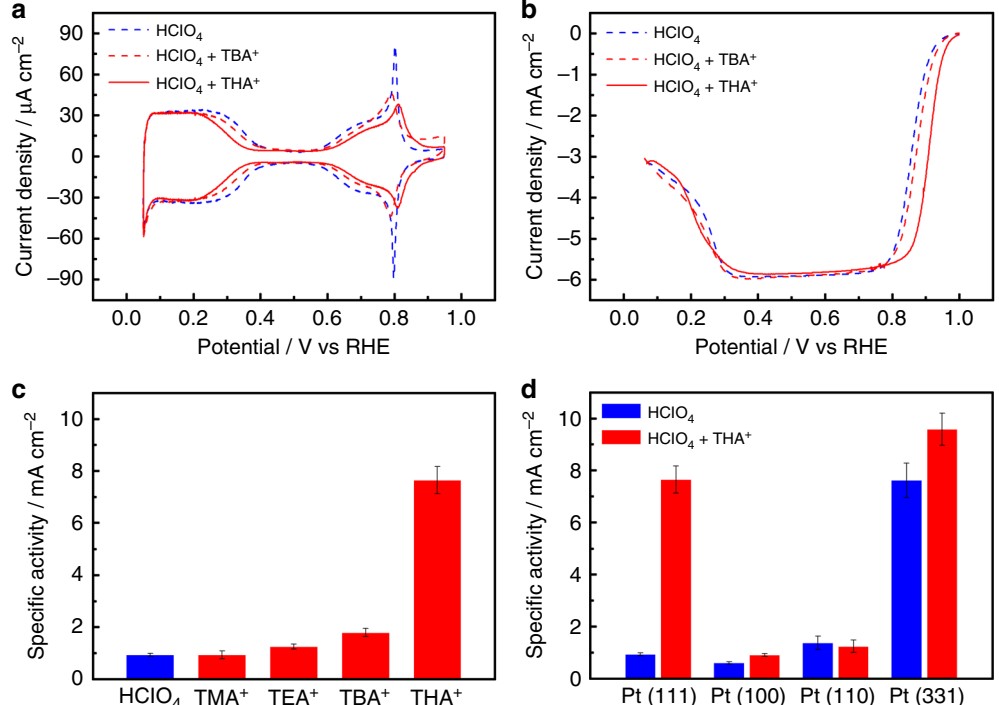

**Fig. 1** Electrochemical characterization of platinum electrodes. **a** Cyclic voltammograms (CVs) and (**b**) linear sweep oxygen reduction reaction (ORR) voltammograms of Pt(111) in 0.1 M HClO$_4$ containing 10$^{-5}$ M tetra-$n$-butylammonium cation (TBA$^+$) and 10$^{-6}$ M tetra-$n$-hexylammonium cation (THA$^+$). CVs were measured in solutions saturated with Ar. The scanning rate is 0.050 V s$^{-1}$. Linear sweep ORR voltammograms were obtained in the solutions saturated with O$_2$, and the potential was scanned from 0.05 V in the positive direction. The scanning rate is 0.010 V s$^{-1}$ and the rotation rate of the electrode is 1600 rpm. **c** Specific ORR activities of Pt(111) in 0.1 M HClO$_4$ containing 10$^{-3}$ M tetramethylammonium cation (TMA$^+$), 10$^{-3}$ M tetraethylammonium cation (TEA$^+$), 10$^{-5}$ M TBA$^+$, and 10$^{-6}$ M THA$^+$ at 0.9 V versus reversible hydrogen electrode (RHE). **d** Specific ORR activities of Pt (111), Pt(100), Pt(110), and Pt(331) = 3(111)–(111) in 0.1 M HClO$_4$ with and without 10$^{-6}$ M THA$^+$ at 0.9 V versus RHE

the OH$_{ad}$ layer. Therefore, we determined the interfacial structures and adsorbed species on Pt(111) in THA$^+$-containing solution in which the degree of activity enhancement was the highest in terms of surface structure and cationic species by using in situ IR spectroscopy and X-ray scattering.

**Infrared spectroscopic measurement of adsorbed species.** Since the absorption frequency of δ$_{PtOH}$ overlaps ν$_{ClO}$ of perchlorate ion, hydrogen fluoride (HF) is used for the IR measurements of OH$_{ad}$. Figure 2a shows the IR spectra of Pt(111) and THA$^+$ modified Pt(111) in 0.1 M HClO$_4$ and 0.1 M HF. The potential-dependent band at 1050 cm$^{-1}$ appears above 0.6 V, and is assigned to the in-plane δ$_{PtOH}$[15,16,20]. Figure 2b shows the potential dependence of the band intensity for δ$_{PtOH}$, and the charge density of Pt oxidation estimated from the CV data (Fig. 1a). The onset potential of OH adsorption shifts positively and the band intensity of δ$_{PtOH}$ at 0.90 V decreases due to the modification with THA$^+$. This result is consistent with the charge density of Pt oxidation in the CV data. The charge density of Pt oxidation increases above 0.80 V, but the band intensity of δ$_{PtOH}$ decreases above 0.80 V. These facts indicate the further oxidation from PtOH to PtO above 0.80 V[14].

The positive- and negative-absorption bands at 1650 and 1610 cm$^{-1}$ are assigned to the HOH bending mode (δ$_{HOH}$) of non-adsorbed and adsorbed hydrogen bonded water, respectively[26,27]. The δ$_{HOH}$ of non-adsorbed water corresponds to those in the liquid phase (1645 cm$^{-1}$) and the solid phase (1650 cm$^{-1}$). It is known that δ$_{HOH}$ shifts to lower frequency by adsorption on metal surfaces. These frequencies are higher than δ$_{HOH}$ for an isolated water monomer (1595 cm$^{-1}$ from matrix isolation

measurements)[28]. Therefore, these water molecules are hydrogen bonded with neighboring oxygen species and electrolyte ions. The generation of OH$_{ad}$ above 0.50 V decreases the coverage of adsorbed water relative to that at the background potential of 0.3 V. The increase in the intensity of δ$_{HOH}$ above 0.5 V also involves the orientation change of water induced by the electrode potential, because the water dipole responds sensitively to the electric field in the EDL. The band intensities of non-adsorbed and adsorbed water are also reduced by the modification with THA$^+$, indicating the decrease of the coverage of OH$_{ad}$. No IR bands in the CH stretching and CH bending regions appear on THA$^+$ modified Pt(111) between 0.30 and 1.0 V, as shown in Supplementary Figure 2. This fact suggests that the structure of the interfacial THA$^+$ is not changed by the electrode potential.

New IR bands appear on THA$^+$-modified Pt(111) at 1510 cm$^{-1}$, which are assigned to the HOH bending mode of adsorbed water monomers at 0.30 V. Since this band overlaps the broad positive-going band around 1570 cm$^{-1}$, an accurate absorption frequency is unclear. The red shift from 1510 cm$^{-1}$ to 1090 cm$^{-1}$ in D$_2$O solvent indicates that these bands are derived from water (Supplementary Figure 4). On Ni(111), Rh(111), and Ru(0001) under ultra-high vacuum condition, the δ$_{HOH}$ shifts from 1595 cm$^{-1}$ (isolated monomer) to 1560 cm$^{-1}$ (adsorbed monomer) by the adsorption via oxygen lone pair[29–31]. The action spectroscopic method using scanning tunneling microscopy also reveals that the signal for water monomer adsorbed on Pt(111) appear around 1550 cm$^{-1}$ [32]. At the solid liquid interface, since the interfacial water molecules are cross-linked via a hydrogen-bonding network, the adsorbed monomers are not a stable species on the electrode. The hydrogen-bonding structure of water in the EDL is influenced by the OHP

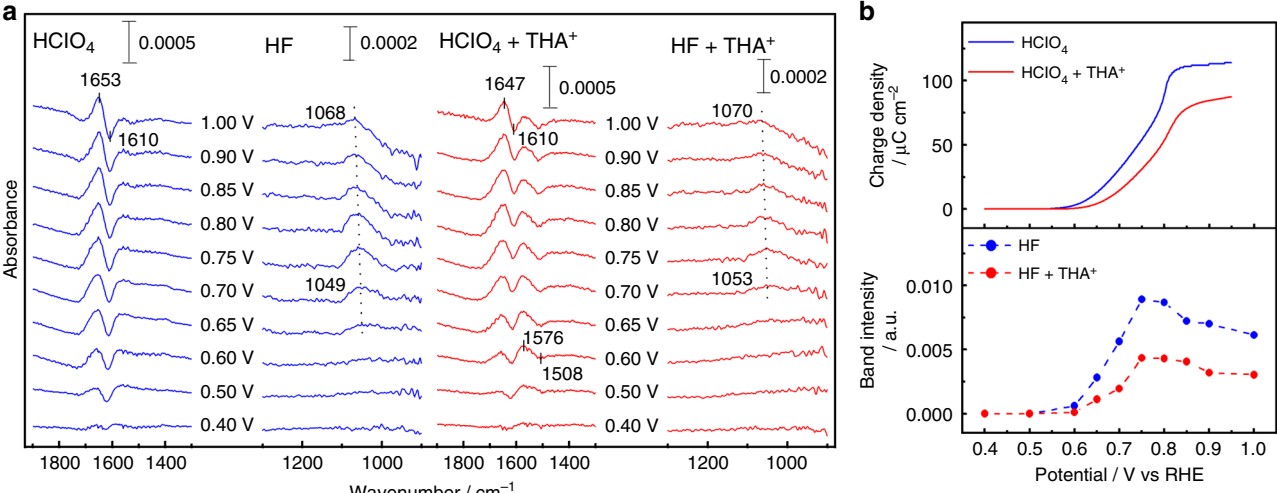

**Fig. 2** Infrared spectroscopic measurement of species adsorbed on platinum(111). **a** Potential dependence of infrared spectra on Pt(111) and Pt(111) modified with tetra-$n$-hexylammonium cation (THA$^+$) in 0.1 M acidic solutions saturated with Ar. The potential of the background spectra is 0.30 V versus reversible hydrogen electrode (RHE). The potentials of the sample spectra are stepped in the positive direction. **b** Potential dependence of the charge density of Pt oxidation and the band intensity of $\delta_{PtOH}$

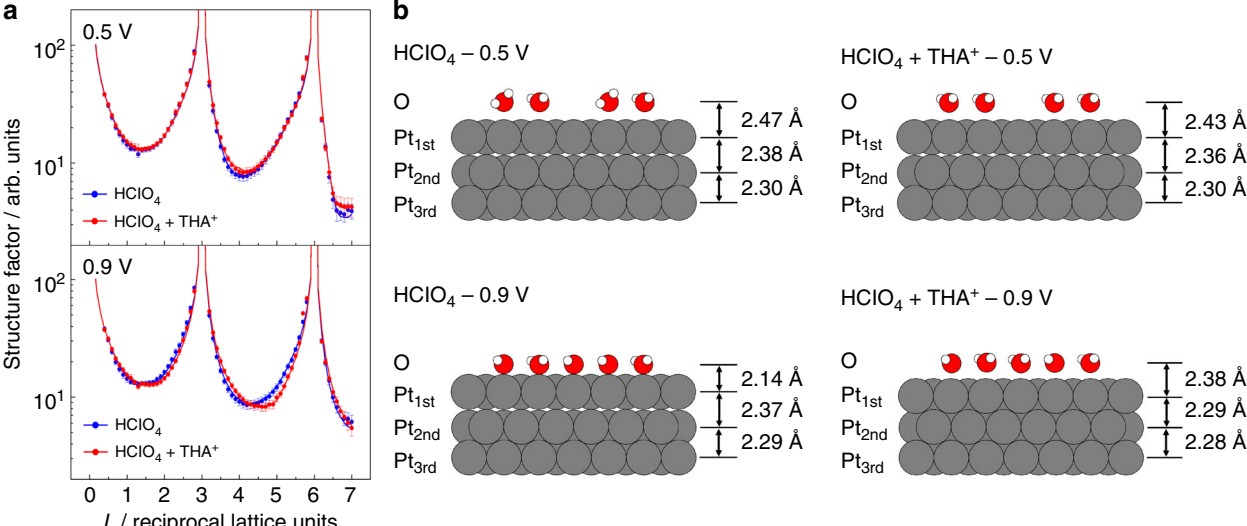

**Fig. 3** X-ray crystal truncation rod measurement of interfacial structure on platinum(111). **a** Specular crystal truncation rod (CTR) profiles of Pt(111) in 0.1 M HClO$_4$ with and without 10$^{-6}$ M tetra-$n$-hexylammonium cation (THA$^+$) saturated with Ar at 0.50 V and 0.90 V versus reversible hydrogen electrode (RHE). The dots are the data points and the solid lines are the structure factors calculated using the optimized model. **b** Schematic models of the interfacial structure

cation and its hydration shell. The hydronium cation facilitates the construction of a tetrahedral configuration, stabilizing the ice-like structure[24,33]. Conversely, the hydration shell around hydrophobic cations such as THA$^+$ disrupts the tetrahedral hydrogen-bonding network. As a consequence, the monomeric species may be stabilized on the surface.

**X-crystal truncation rod measurement of interfacial structures.** Electron density profiles along the surface normal direction were estimated from X-ray specular crystal truncation rod (CTR) scattering. Figure 3a shows the specular CTR profiles of Pt(111) in 0.1 M HClO$_4$ with and without 10$^{-6}$ M THA$^+$. The CTR profiles were obtained at 0.5 and 0.9 V, which are the double layer and the OH adsorption regions, respectively. Normalized CTR profiles are shown in Supplementary Figure 5a. While the CTR

profiles with and without THA$^+$ at 0.50 V are nearly identical, the CTR at 0.90 V is clearly altered by the addition of THA$^+$. Since THA$^+$ is composed of nitrogen, carbon, and hydrogen, the electron density in the EDL is significantly lower than that of Pt. Therefore, the primary factor for this variation in the CTR is the structural change in the substrate Pt. The initial model used for structural optimization comprises three Pt layers and one oxygen layer of water molecule or OH$_{ad}$ layer. The vertical atomic position, occupancy factor, and Debye–Waller factors for the Pt and oxygen layers were optimized by the least-squares method. The electron density profiles and the structural parameters of the optimized model are shown in Supplementary Figure 5b and Supplementary Table 1, respectively.

Schematic models of the interfacial structure are shown in Fig. 3b. At 0.50 V, the layer distance between oxygen and the 1st

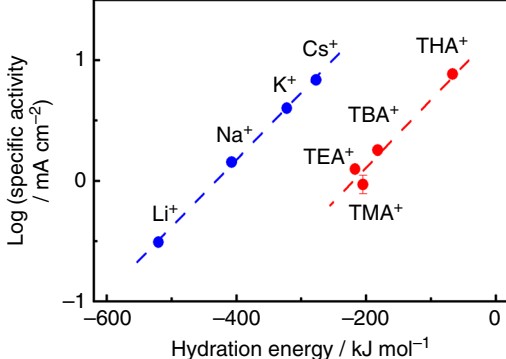

**Fig. 4** Specific oxygen reduction reaction activities and hydration energies of cations. Logarithm of the specific oxygen reduction reaction (ORR) activities versus hydration energies of alkali metal cations[19], tetramethylammonium cation (TMA+), tetraethylammonium cation (TEA+), tetra-n-butylammonium cation (TBA+), and tetra-n-hexylammonium cation (THA+). The hydration energies of tetraalkylammonium cations were calculated using a following equation: $\Delta_{hydr}H°([H(CH_2)_n]_4N^+) = -172-42.4n + 10n^2$ [25]

Pt ($d_{O-1stPt}$) is ~2.43–2.47 Å in both solutions. Therefore, the oxygen species are assigned to the adsorbed water at the on-top site on Pt(111). A similar bond length between water oxygen and Pt is proposed by X-ray measurements and DFT calculations[20,34–36]. The layer distance between 1st and 2nd Pt ($d_{1stPt-2ndPt} = 2.36$–2.38 Å) expands by 4–5% of the bulk layer ($d_{bulk} = 2.27$ Å). This surface relaxation is caused by charge transfer from the adsorbed water to Pt and the electric field[37]. The coverage of oxygen species ($\theta_O$) slightly increases for the THA+-containing solution. IR measurements of interfacial water have been performed by many researchers, revealing that the orientation of the adsorbed water and the hydrogen-bonding networks are altered by the interaction with the electrolyte ions[38,39]. The increase of the coverage indicates the existence of THA+ at the interface. The IR spectra in Fig. 2a show that OH_ad is not adsorbed at 0.50 V. However, the negative-going band at 1610 cm$^{-1}$ for adsorbed water on Pt(111) without THA+ is larger than that with THA+. This fact indicates that the coverage of adsorbed water $\theta_O$ on Pt(111) with THA+ is higher than that without THA+ in 0.1 M HClO_4.

At 0.9 V, The $d_{O-1stPt}$ in HClO_4 solution decreases from 2.47 Å to 2.14 Å. This corresponds to the Pt-O bond length of OH_ad on Pt(111) proposed by low-energy electron diffraction and in situ X-ray diffraction measurements[34,40]. Potential-dependent coadsorbed structures of water and OH_ad were suggested by DFT calculations and Monto Carlo simulations[41,42]. XPS measurements also indicate the coadsorption of water and OH_ad[14], hence the coverage at 0.90 V ($\theta_O = 0.78$) includes adsorbed water and OH species. The surface relaxation of the Pt layer is promoted by electron donation via an oxygen lone pair of the OH_ad as well as adsorbed water[43]. Therefore, $d_{1stPt-2ndPt}$ is comparable with that at 0.50 V. Conversely, in the presence of THA+, $d_{O-1stPt}$ is 2.38 Å, which indicates that adsorbed water is dominant instead of OH_ad. It is notable that the surface relaxation is eliminated ($d_{1stPt-2ndPt} = 2.29$ Å) even though the water and OH_ad are adsorbed on Pt. This result indicates that the interaction between the oxygen species and Pt is decreased by the presence of THA+.

## Discussion

The enhancement of the ORR activity by the addition of TAA+ is related with the hydrophobicity of TAA+. Figure 4 shows the correlation between hydration enthalpy and the ORR activity. A linear correlation is found except TMA+, and the slope for TAA+

is similar to that for alkali metal cations, as reported by Markovic et al.[19]. They suggested that the non-covalent interaction between hydrated cations and OH_ad affects electrocatalytic reaction on Pt[19,44,45]. In alkaline solution, the ORR activity is related to the hydration energy of alkali metal cation because cations with a strong affinity for adsorbed oxygen species block the active sites for the ORR. The order of the hydration energy of alkali metal cations (Li+ ≫; Na+ > K+ > Cs+) is inversely correlated to the ORR activity (Cs+ > K+ > Na+ ≫ Li+). Previous IR and X-ray studies have revealed that Li+ located at the OHP inhibits the surface oxidation of Pt(111) due to the stabilization effect between Li+ and OH_ad, whereas high-order oxidation accompanied by surface roughness proceeds in solution containing Cs+ [20]. Thus, OH_ad is stabilized by the interaction with species that have high-oxygen affinity. The hydration structures of interfacial cations were assumed from the OH stretching band frequency[21,22]. The hydration water around hydrophilic cations, such as H+ and Li+, is coordinated with the dipole moment pointed outward and the hydrogen atoms of the hydration water, which can link with the oxygen species located in the secondary hydration sphere. Conversely, hydrophobic cations such as TAA+ weaken the bonding with hydration water and strengthen intermolecular hydrogen bonding in the primary hydration shell. Consequently, the hydration shell around the hydrophobic cation is restricted to coordination with the outer oxygen species because of the complete hydrogen-bonding network within the shell. The ORR activity in TMA+ is comparable with that in HClO_4 without TAA+, indicating that TMA+ does not contribute the enhancement of the ORR. Since the hydration shell size of TMA+ is smaller than those of the other TAA+, the complete hydrogen-bonding network may not be constructed in the shell.

The OHP cations interact with the charged species in the EDL as well as the substrate. Therefore, cations electrostatically interacting with adsorbed anions can approach the surface at the potentials more positive than the potential of zero charge (pzc)[17,20,46,47]. The coverage of the OHP cations is balanced with that of the adsorbed anions and the surface charge depending on the electrode potential. On the surface at a constant coverage of adsorbed counter anions, the coverage of the OHP cation decreases with increasing potential[17]. As described above, the IR spectra indicate that the coverage of THA+ is approximately constant in the potential region examined in this study. The increase in the coverage of the negatively charged species, i.e., OH_ad and O_ad, is balanced by positive surface charge so that the coverage of the THA+ remains constant.

While the ORR activity on Pt(111) is dramatically enhanced by the presence of THA+, there is no significant enhancement of the ORR on Pt(100), Pt(110) and Pt(331) by THA+. In acidic solution above 0.6 V, the hydration water around H+ is linked with OH_ad through hydrogen bond (Fig. 5a). H+–H_2O–OH_ad formation stabilizes the OH_ad layer and inhibits the ORR. According to studies on the Pt(111) surface under the ultra-high vacuum condition, the coadsorption of OH_ad and H_2O forms a well-defined 3 × 3 honeycomb structure because the symmetry and OH···O distance in the coadsorbed layer fit well with the Pt(111) lattice[14,48]. The approach of the hydration shell around THA+ to the Pt surface disrupts the stabilization effect between the hydration water and the OH_ad layer (Fig. 5b). The coverage of OH_ad is decreased by this destabilization effect, which induces the disruption of stable hydrogen-bonding network in the coadsorbed layer. These multiple effects are enhanced by the hydrophobicity of cation and promote the access of oxygen molecules to the Pt(111) surface.

The IR spectra on Pt(100) indicate that the addition of THA+ results in a slight reduction of the band intensity and band broadening of $\delta_{PtOH}$ as shown in Supplementary Figure 6. The charge densities for surface oxidation also decrease on Pt(110) and Pt(331) above 0.7 V after the addition of THA+. For Pt(100),

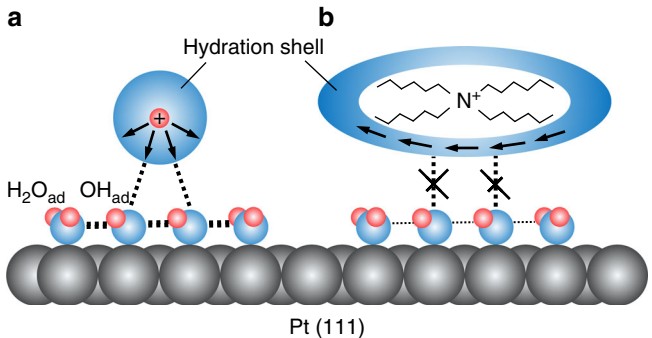

**Fig. 5** Schematic of interfacial hydrated cations and adsorbed hydroxide species. **a** hydrophilic cation and (**b**) hydrophobic cation. Arrows indicate the direction of water dipole within the hydration shell. (Platinum is represented by gray spheres, oxygen by blue spheres, and hydrogen by pink spheres.)

Pt(110), and Pt(331), although OH adsorption is inhibited by the approach of the THA$^+$ hydration shell, the ORR enhancement effect by the disruption of OH$_{ad}$ + H$_2$O hydrogen-bonding network is weak due to the narrow (111) terrace and the lattice mismatch of the OH$_{ad}$ + H$_2$O honeycomb layer with the substrate. The Pt(331) surface having a (111) terrace shows high-ORR activity in HClO$_4$ without TAA$^+$ [8]. DFT calculations suggest that the OH$_{ad}$ on stepped Pt surfaces is destabilized by deforming the hydrogen bonds of interfacial water surrounding the adsorption sites, and the destabilization of oxygen species cause the enhancement of the ORR on the (111) terrace[49].

The ORR activity on Pt(111) in THA$^+$ is comparable with those on Pt$_3$Co(111) and Pt$_3$Ni(111) in HClO$_4$. The activation factor for the ORR with these Pt alloys is also due to decrease in the coverage of OH$_{ad}$. Therefore, a similar inhibiting effect of OH$_{ad}$ can be achieved by the structural improvement in EDL species. Furthermore, the supply of oxygen molecules to the electrode may be improved by weakening the hydrogen-bonding strength around the hydrophobic cations.

In summary, we evaluated the ORR activities of single-crystal Pt electrodes in HClO$_4$ solutions containing TAA$^+$ with various alkyl chain lengths. The ORR activity increased in the following sequence THA$^+$ $n = 6$ » TBA$^+$ $n = 4$ > TEA$^+$ $n = 2$ > TMA$^+$ $n = 1$ ≈ HClO$_4$, and the ORR activity on Pt(111) in THA$^+$ was comparable with those on Pt$_3$Co(111) and Pt$_3$Ni(111) in HClO$_4$. In situ IR and X-ray CTR measurements indicated that the coverage of OH$_{ad}$ on Pt(111) at 0.90 V in the presence of THA$^+$ is decreased. The weaker interaction between the hydration shell around THA$^+$ and OH$_{ad}$ destabilizes the OH$_{ad}$ + H$_2$O coadsobed layer and forms a highly efficient interface for the ORR.

## Methods

**Electrochemical measurement**. Single-crystal Pt beads for CV measurement were prepared by the Clavilier's method[50]. The crystal beads were oriented to (111), (100), (110), and (331) using Laue back reflection of X-ray, and then mechanically polished with diamond slurries[51]. A Pt(111) disk (8 mm in diameter, MaTech) was used for the X-ray CTR measurement. The samples were annealed in H$_2$ + O$_2$ flame, and then cooled to room temperature in Ar + H$_2$ or Ar atmosphere. The annealed surfaces were protected with ultrapure water (Milli–Q Advantage A10, Millipore). The solutions were prepared with HClO$_4$ (Kanto Chemical), HF (Kanto Chemical), tetraalkylammonium salts, and ultrapure water. The tetraalkylammonium salts used were tetramethylammonium perchlorate (Tokyo Chemical Industry), tetraethylammonium perchlorate (Tokyo Chemical Industry), tetra-$n$-butylammonium perchlorate (Sigma-Aldrich), and tetra-$n$-hexylammonium perchlorate (Alfa Aesar). The reference electrode used in all measurements was a reversible hydrogen electrode (RHE). The ORR voltammograms were measured in the hanging meniscus rotating disk electrode (RDE) configuration using an electrochemical analyzer (ALS 701DH, BAS) and a rotating ring-disk electrode apparatus (RRDE-3 A, BAS)[52]. The ORR activity was estimated from the kinetic current density (specific activity, $j_k$) at 0.90 V versus RHE according to the Koutecky–Levich equation: $1/j = 1/j_k + 1/j_L$, where $j$ and $j_L$ are the total current density

and the limiting current density, respectively. The current densities were normalized to the geometrical surface area of the Pt surface.

**Infrared spectroscopy**. In situ infrared reflection absorption spectroscopy (IRAS) measurement was performed using an Fourier transform (FTIR) spectrometer (Bruker Vertex 70 v). An electrochemical IR cell with a trapezoid BaF$_2$ or CaF$_2$ window beveled at 60° was attached to the spectrometer with a narrow band mercury–cadmium–telluride (MCT) detector. A polypropylene film (Chemplex Industries) with a thickness of 6 μm was intercalated between the BaF$_2$ prism and the Pt(111) and Pt(100) surfaces to prevent the dissolution of the BaF$_2$ material[20]. A layer of water was placed between the BaF$_2$ prism and the polymer film to compensate for the gap of the refractive index. The interferograms were averaged over 640 scans by subtractively normalized interfacial FTIR (SNIFTIRS) with p-polarized light at a resolution of 4 cm$^{-1}$. In THA$^+$-containing solution, the adsorption of OH$_{ad}$ is inhibited by the repetitive potential steps of SNIFTIRS measurement because THA$^+$ assembles on the surface at the negative potentials. Therefore, after the Pt(111) and Pt(100) electrodes were immersed in 0.1 M HClO$_4$ and 0.1 M HF containing THA$^+$ for 5 min, IR measurement was performed in the solution without THA$^+$. We confirmed that Pt(111) modified with THA$^+$ shows the same voltammograms and ORR activity.

**X-ray crystal truncation rod scattering**. In situ X-ray CTR measurement was performed with a multi-axis diffractometer at BL13XU (SPring-8)[53]. The X-ray energy used was 20 keV. Integrated intensities were measured by rocking scans, and then corrected for Lorentz and area factors. Specular CTR was measured along the surface normal direction $L$ in units of $\mathbf{c^*} = 2\pi/\mathbf{c}$, $|\mathbf{c}| = 0.6797$ nm. Structure refinements were conducted using the least-squares method with the ANA-ROD program[54].

## Data availability

The data that support the findings of this study are available within the paper and Supplementary Information, as well as from the corresponding author upon request.

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

## Acknowledgements

X-ray measurements were supported by JASRI/SPring−8 (2016A1287 and 2017B1293). This work was supported by the Asahi Glass Foundation, JSPS KAKENHI Grant Number 2435001 and 15H03763, and NEDO.

## Author contributions

M.N. designed the study. T.K. performed the electrochemical measurements. M.N. and T.K. performed the IR measurements. M.N., T.K., H.T. and O.S. performed the X-ray CTRs measurements. M.N., T.K. and N.H. discussed the Results. M.N. and T.K. co-wrote the paper with contributions from all authors.
