## [Peer Review File · Nature Communications]

Reviewers' comments:

Reviewer #1 (Remarks to the Author):

Review on "Effect of hydrophobic cations on oxygen reduction reaction on single crystal electrodes of Pt" by Kumeda et al.

The manuscript describes an improved oxygen reduction reaction (ORR) in a combined measurements using rotating-disk electrode (RDD), FTIR, surface x-ray scattering. The RDD scans show that the onset potential shift by ~300 mV. This is a significant shift and an interesting results, if things make sense.

The authors' main idea is adding hydrophobic TAA+ to the solution to disrupt the adsorbed OH layer. One immediate question is then TAA+ has little effect but THA+ has a significant effect. What is the difference in hydrophobicity strength between them? Is the hydrophobicity correlated to ORR? Authors seem suggesting both are hydrophobic but yet two completely different effect on ORR of Pt(111).

THA+ is to disrupt the OH layer, which known to be a blocking species for ORR. However, according to Fig.1a and Fig.2b, the changes in the charges in the butterfly wing area is not very large. Also, Fig. 1a does not agree well with RDD in Fig.1b. The butterfly wing for TBA+ shows small negative shift in Fig.1a but the ORR current shows a positive shift in Fig.1b. THA+ shows a small positive shift in Fig.1a, but a rather large 300 mV shift in Fig.1b. In Fig.1a certainly shows no significant decrease in the areas under the butterfly wings to justify the 300 mV shift. All I can see is that the long-range ordering, signified by the sharp spikes at 800 mV, is diminished. The diminishing spikes is well known to be typical for impurity adsorption. It makes no sense anyway why + ions have any effect on ORR unless it introduces some impurities or THA+ chemisorbed at negative potential. Positive ions will be pushed away from the surface as soon as the potential crosses the pzc. The ORR potential is definitely far more positive than the pzc. Therefore, Fig. 4b makes no sense.

Another argument against disruption of the OH layer is the results for other surfaces. OH adsorption occurs on all surfaces, (100), (110), and (331). If so, why there is no effect on those surfaces. The higher ORR at the high index surface such as (331), which is a combination of (333) and (0 0 -2) facets, has well been established, simply based on geometric arguments of ORR on nanofacets. For the authors' OH argument to be valid, similar effects should exist for all surfaces.

FTIR measurements are sometimes up to the interpretations. In Fig.2, what does "The reference potential is 0.30 V vs RHE." mean? Does it mean that the reference electrode was Ag/AgCl? The reference electrodes used in the studies must be clearly specified. Also, authors should show the spectra on positive and negative sweeps.

The surface x-ray scattering (SXS) measurements are interesting. However, note that SXS has not been measured at 1.0V, the starting potential for RDD, where the place-exchange oxidation should have occurred in Pt(111). Whatever happens to the surface by adding THA+ may simply suppress the place-exchange oxidation, thereby, affecting the ORR. Still, the suppression is likely by some unknown impurities or chemisorption, unlikely THA+ that will be well pushed away from the surface at the ORR potential. In fitting the data with small differences, it is better to fit the difference counts between the potentials at every potential. It is not all clear that the blue for 0.5 V and the blue for 0.9V are identical. They look different, although they should be identical. Since they are not shown together, it is difficult to judge, though. If the sample was moved between these four measurements, it will be difficult to quantify the differences for such small differences in scattering.

Reviewer #2 (Remarks to the Author):

The authors of this manuscript reported an interesting cation effect on ORR activity of Pt single crystal electrodes. By adding tetraalkylammonium perchlorates in HClO₄ solutions, they observed significant change in ORR activity, with a maximum eightfold enhancement at 0.9 V for Pt(111). Although the method may not be applicable for nanocatalysts in fuel cells, the origin of this activity enhancement should be interesting to the researchers in electrocatalysis field.

As authors discussed, it is known that alkali metal cations (from Li⁺ to Cs⁺) affect the stability of OH_{ad}, and thus, the ORR activity because lowering the coverage of OH_{ad} at high potential holds the key for enhancing the ORR activity on Pt. This manuscript appears to be the first paper by my knowledge that clearly demonstrated an ORR enhancement by hydrophobic cations via their effect on the OH_{ad} stability.

To identify the origin of the activity enhancement, the authors carried out IR spectroscopy and X-ray scattering measurements for Pt(111) in HClO₄ without and with one of the most effective cation. The results showed that the hydrophobic cations are not adsorbed at the Pt electrode surfaces, but their presence lowered the OH coverage around 0.9 V. The data analyses and discussion about hydrogen bonding are mostly convincing.

However, the authors should discuss why the cation effect strongly depends on lattice facet even though neither the cations or OH_{ad} for ordered structures on single crystal surfaces. Based on the arguments for enhanced ORR activity on (111), should one expect stronger OH adsorption on Pt(100) with or without hydrophobic cation than Pt(111)? If so, could confirmation be made by adding IR results for Pt(100).

I recommend publication of the manuscript after a revision. A minor suggestion is to change X-ray diffraction to X-ray scattering as the specular CTR is considered as a X-ray scattering technique.

Reviewer #3 (Remarks to the Author):

In this manuscript, the authors have discovered that the co-adsorption of hydrophobic cations such as tetraalkylammonium cations (TAA⁺) greatly enhances the oxygen reduction reaction (ORR) of Pt electrodes. The activity of ORR increases by elongating the alkyl chains of TAA⁺ from methyl to hexyl, and the activity also depends on the surface structure of Pt electrodes. Activity of ORR is very important to increase the efficiency of the fuel cells; therefore I think this paper attracts considerable attentions of the readers of this journal. Before the publication, following comments should be considered. These would be useful to improve this manuscript.

1. In Figure 1c, the authors found THA⁺ shows highest activity for the ORR. However, the detailed mechanism is not written in the manuscript. The effects of the alkyl chains of TAA⁺ should be discussed.

2. The effects of the surface structure (Fig. 1d) of Pt are not discussed. The mechanism why THA⁺ is effective only for Pt(111) should be discussed.

3. In Fig. 2, the authors are describing about the intensity changes of the bands at 1650 and 1620 cm⁻¹, and proposed OH(ad) are formed above 0.5 V. Hydrogen-bonding networks at the electric-double layers are very sensitive to the orientation of adsorbed water molecules. The orientation of water depends on the applied potentials; therefore the effects of the electric fields should be also discussed.

4. In page 7, it seems that the concentration of THA⁺ on Pt electrodes does not change from 0.3 to 1.0 V. THA⁺ is a positively charged molecule, why did not change ?

To reviewers

Thank you for the valuable comments for the improvement of our manuscript. The manuscript has been revised according to the comments. The revised parts are painted in yellow in the manuscript.

Reviewer 1

Comment 1

The authors' main idea is adding hydrophobic TAA⁺ to the solution to disrupt the adsorbed OH layer. One immediate question is then TAA⁺ has little effect but THA⁺ has a significant effect. What is the difference in hydrophobicity strength between them? Is the hydrophobicity correlated to ORR? Authors seem suggesting both are hydrophobic but yet two completely different effect on ORR of Pt(111).

Reply

The coordination energies of hydration water with TAA cations depends on the alkyl chain length of TAA⁺. As shown in Fig. 4, we found the strong relationship between the hydration enthalpy estimated from Ref.25 and the ORR activity except TMA⁺. Furthermore, this trend corresponds to the relationship between the ORR activity and the hydration energy of alkaline metal cations in alkaline solutions (Ref. 19). Thus, the hydrophobicity strength depending on the alkyl chain length is one of the enhancement factors for the ORR activity. We also mentioned the reason the deviation of TMA⁺. Following sentences were added in the **Discussion** and the relationship between the hydration energy and the ORR activity was added in **Figure 4**.

“The enhancement of the ORR activity by the addition of TAA⁺ is related with the hydrophobicity of TAA⁺. Figure 4 shows the correlation between hydration enthalpy and the ORR activity. A linear correlation is found except TMA⁺ and the slope for TAA⁺s is similar to that for alkali metal cations, as reported by Markovic et al¹⁹.”

“The ORR activity in TMA⁺ is comparable to that in HClO₄ without TAA⁺, indicating that TMA⁺ does not contribute the enhancement of the ORR. Since the hydration shell size of TMA⁺ is smaller than those of the other TAA⁺s, the complete hydrogen-bonding network may not be constructed in the shell.”

Comment 2

THA⁺ is to disrupt the OH layer, which known to be a blocking species for ORR. However, according to Fig.1a and Fig.2b, the changes in the charges in the butterfly wing area is not very large. Also, Fig. 1a does not agree well with RDD in Fig.1b. The butterfly wing for TBA⁺ shows small negative shift in Fig.1a but the ORR current shows a positive shift in Fig.1b. THA⁺ shows a small positive shift in

Fig.1a, but a rather large 300 mV shift in Fig.1b. In Fig.1a certainly shows no significant decrease in the areas under the butterfly wings to justify the 300 mV shift. All I can see is that the long-range ordering, signified by the sharp spikes at 800 mV, is diminished. The diminishing spikes is well known to be typical for impurity adsorption.

Reply

It is known that the OH_{ad} species is deactivation species for the ORR and the OH_{ad} is stabilized by the interaction with hydrophilic cation. However, on Pt(111), a stable hydrogen bonding network of $\text{OH}_{\text{ad}} + \text{H}_2\text{O}$ coadsorbed layer also inhibits the ORR for the following reasons. Study on n(111)–(111) Pt surfaces shows that the (111) terrace edge enhances the ORR activity. DFT calculations predict that the (111) terrace edge changes the structure of hydrogen bonding network on n(111)–(111) Pt surfaces. The structural change of the coadsorbed layer hinders the formation of Pt oxides that are known to deactivate the ORR, resulting in a higher ORR activity than Pt(111) electrodes without a terrace edge. The approach of the hydrophobic cation destabilizes the OH_{ad} layer, which causes the deformation of the stable hydrogen bonding network on Pt(111). Similar activation effect with n(111)–(111) is expected to occur on Pt(111). IR spectra of adsorbed water also suggest the change in hydrogen bonding structure. Following sentences were added in the **Discussion**.

“While the ORR activity on Pt(111) is dramatically enhanced by the presence of THA^+ , there is no significant enhancement of the ORR on Pt(100), Pt(110) and Pt(331) by THA^+ . In acidic solution above 0.6 V, the hydration water around H^+ is linked with OH_{ad} through hydrogen bond (Fig. 5a). $\text{H}^+ - \text{H}_2\text{O} - \text{OH}_{\text{ad}}$ formation stabilizes the OH_{ad} layer and inhibits the ORR. According to studies on the Pt(111) surface under the ultra-high vacuum condition, the coadsorption of OH_{ad} and H_2O forms a well-defined 3×3 honeycomb structure because the symmetry and $\text{OH} \cdots \text{O}$ distance in the coadsorbed layer fit well with the Pt(111) lattice^{14,48}. The approach of the hydration shell around THA^+ to the Pt surface disrupts the stabilization effect between the hydration water and the OH_{ad} layer (Fig. 5b). The coverage of OH_{ad} is decreased by this destabilization effect, which induces the disruption of stable hydrogen bonding network in the coadsorbed layer. These multiple effects are enhanced by the hydrophobicity of cation and promote the access of oxygen molecules to the Pt(111) surface.”

As you have mentioned, the spikes at 0.8 V shift negatively in TBA^+ despite the positive shift in THA^+ . The reason of the potential shift of spike peaks is unclear, but the onset potential ($\text{THA}^+ > \text{TBA}^+ > \text{HClO}_4$) and total charge density between 0.5 V and 0.9 V ($\text{THA}^+ < \text{TBA}^+ < \text{HClO}_4$) correlates to the ORR activity. Feliu et al. suggested that the broad and sharp peaks in the butterfly wing are due to the dissociative adsorption of two different kinds of interfacial water (Electrochem. Commun, 9,

2789 (2007). This indicates that the sharp spikes are diminished by the structural change of interfacial water by the hydrophobic cation.

Comment 3

It makes no sense anyway why + ions have any effect on ORR unless it introduces some impurities or THA^+ chemisorbed at negative potential. Positive ions will be pushed away from the surface as soon as the potential crosses the pzc. The ORR potential is definitely far more positive than the pzc. Therefore, Fig. 4b makes no sense.

Reply

Recent X-ray scattering studies reveal the detailed electrical double layer including the OHP cation. On Ag(100) in CsBr solution (pzc ~ -0.9 V vs Ag/AgCl), the Cs cation forms the OHP layer above c(2x2)-Br layer between -0.6 V and -0.03 V vs Ag/AgCl. Similar OHP cation layer at positive potential was reported on Ag(111) in KOH, Cu(100) in $\text{H}_2\text{SO}_4+\text{KCl}$, and Pt(111) in alkaline solutions. These studies indicate that the OHP cation is located at the potential more positive than the pzc because of the non-covalent interactions between adsorbed anion (partially negatively charged) and the OHP cation. The amount of the OHP cation is balanced with that of adsorbed anion and the surface charge in the electrical double layer. Following sentences were added in the **Discussion**.

“The OHP cations interact with the charged species in the EDL as well as the substrate. Therefore, cations electrostatically interacting with adsorbed anions can approach the surface at the potentials more positive than the potential of zero charge (pzc)^{17,20,46,47}. The coverage of the OHP cations is balanced with that of the adsorbed anions and the surface charge depending on the electrode potential. On the surface at a constant coverage of adsorbed counter anions, the coverage of the OHP cation decreases with increasing potential¹⁷. As described above, the IR spectra indicate that the coverage of THA^+ is approximately constant in the potential region examined in this study. However, the coverage of the negatively charged species, i.e. OH_{ad} and O_{ad} , increases with increasing potential, which compensates for the decrease of the THA^+ .”

Comment 4

Another argument against disruption of the OH layer is the results for other surfaces. OH adsorption occurs on all surfaces, (100), (110), and (331). If so, why there is no effect on those surfaces. The higher ORR at the high index surface such as (331), which is a combination of (333) and (0 0 -2) facets, has well been established, simply based on geometric arguments of ORR on nanofacets. For the authors' OH argument to be valid, similar effects should exist for all surfaces.

Reply

As described in the reply against the comment 2 of review 1, the ORR activity depends on the stability of hydrogen bonding network as well as the OH_{ad} coverage. On the (111) terrace, the former factor is dominant and the ORR is significantly enhanced on Pt(111). The (111) terrace in (110) and (331) is not wide enough to form the hydrogen bonded honeycomb structure, and the 4-fold symmetry on Pt(100) does not fit in well with the honeycomb structure. We previously reported the band intensity of the δ_{PtOH} vs the ORR activity on n(111)-(111) Pt surfaces. Pt(110) differs significantly from the linear trend of the other surfaces. Reconstructed Pt(110) shows an anomalously low ORR activity for the n(111)-(111) series. The reaction mechanism of the ORR on Pt(110) may be different from that of other n(111)-(111) surfaces. Following sentences were added in the **Discussion**.

“The IR spectra on Pt(100) indicate that the addition of THA^+ results in a slight reduction of the band intensity and band broadening of δ_{PtOH} as shown in Fig. S6. The charge densities for surface oxidation also decrease on Pt(110) and Pt(331) above 0.7 V after the addition of THA^+ . For Pt(100), Pt(110), and Pt(331), although OH adsorption is inhibited by the approach of the THA^+ hydration shell, the ORR enhancement effect by the disruption of $\text{OH}_{\text{ad}} + \text{H}_2\text{O}$ hydrogen bonding network is weak due to the narrow (111) terrace and the lattice mismatch of the $\text{OH}_{\text{ad}} + \text{H}_2\text{O}$ honeycomb layer with the substrate. The Pt(331) surface having a (111) terrace shows high ORR activity in HClO_4 without TAA^{+8} . DFT calculations suggest that the OH_{ad} on stepped Pt surfaces is destabilized by deforming the hydrogen bonds of interfacial water surrounding the adsorption sites and the destabilization of oxygen species cause the enhancement of the ORR on the (111) terrace⁴⁹.”

Comment 5

FTIR measurements are sometimes up to the interpretations. In Fig.2, what does “The reference potential is 0.30 V vs RHE.” mean? Does it mean that the reference electrode was Ag/AgCl? The reference electrodes used in the studies must be clearly specified. Also, authors should show the spectra on positive and negative sweeps.

Reply

The reference electrode used in this work is reversible hydrogen electrode (RHE). In the caption of Fig. 2, “The reference potential is 0.30 V vs RHE” means that the measured potential of the background spectrum is 0.30 V vs RHE. Following sentences were added in the captions of **Figures 1 and 2**.

“Linear sweep ORR voltammograms were obtained in the solutions saturated with O_2 and the potential was scanned from 0.05 V in the positive direction.” and “The potential of the background spectra is 0.30 V vs RHE. The potentials of the sample spectra are stepped in the positive direction.”

The linear sweep voltammogram of Fig. 1b is scanned from 0.05 V to 1.0 V. Therefore, for the IR measurements, the potentials of sample spectra are stepped in the positive direction from 0.30 V to 1.0 V. Although the spectra of Fig. 2 were obtained using SNIFTIRS method in order to achieve a high S/N ratio, similar spectra were also observed by conventional stepwise method using the potential sweep to anodic or cathodic direction.

Comment 6

The surface x-ray scattering (SXS) measurements are interesting. However, note that SXS has not been measured at 1.0V, the starting potential for RDD, where the place-exchange oxidation should have occurred in Pt(111). Whatever happens to the surface by adding THA^+ may simply suppress the place-exchange oxidation, thereby, affecting the ORR. Still, the suppression is likely by some unknown impurities or chemisorption, unlikely THA^+ that will be well pushed away from the surface at the ORR potential. In fitting the data with small differences, it is better to fit the difference counts between the potentials at every potential. It is not all clear that the blue for 0.5 V and the blue for 0.9V are identical. They look different, although they should be identical. Since they are not shown together, it is difficult to judge, though. If the sample was moved between these four measurements, it will be difficult to quantify the differences for such small differences in scattering.

Reply

In the estimation of the ORR activity using the linear sweep voltammetry, the ORR activity depends on the holding time at 1.0 V vs RHE because of the place-exchange oxidation as you mentioned (Tanaka et al. *Electrocatalysis*, 5, 354 (2014)). In this study, the linear sweep ORR voltammograms were performed by positive sweep from 0.05 V to 1.0 V. Therefore, we do not discuss the interfacial structure at 1.0 V. In case that the starting potential is 1.0 V, the discussion of the structure at 1.0 V will be necessary. The differential intensity profiles along the CTR between 0.5 V and 0.9 V was added in **Fig. S5**.

Reviewer 2

Comment 1

The authors should discuss why the cation effect strongly depends on lattice facet even through neither the cations or OH_{ad} for ordered structures on single crystal surfaces. Based on the arguments for enhanced ORR activity on (111), should one expect stronger OH adsorption on Pt(100) with or without hydrophobic cation than Pt(111)? If so, could confirmation be made by adding IR results for Pt(100).

Reply

It is known that the OH_{ad} species is deactivation species for the ORR and the OH_{ad} is stabilized by the interaction with hydrophilic cation. However, on Pt(111), a stable hydrogen bonding network of OH_{ad} + H₂O coadsorbed layer also inhibits the ORR. Thus, the ORR activity depends on the stability of hydrogen bonding network as well as the OH_{ad} coverage. On the (111) terrace, the former factor is dominant and the ORR is significantly enhanced on Pt(111) for the following reasons. Study on n(111)–(111) Pt surfaces shows that the (111) terrace edge enhances the ORR activity. DFT calculations predict that the (111) terrace edge changes the structure of hydrogen bonding network on n(111)–(111) Pt surfaces. The structural change of the coadsorbed layer hinders the formation of Pt oxides that are known to deactivate the ORR, resulting in a higher ORR activity than Pt(111) electrodes without a terrace edge. The approach of the hydrophobic cation destabilizes the OH_{ad} layer, which causes the deformation of the stable hydrogen bonding network on Pt(111). Similar activation effect with n(111)–(111) is expected to occur on Pt(111). IR spectra of adsorbed water also suggest the change in hydrogen bonding structure.

The (111) terrace in (110) and (331) is not wide enough to form the hydrogen bonded honeycomb structure, and the 4-fold symmetry on Pt(100) does not fit in well with the honeycomb structure. We previously reported the band intensity of the δ_{PtOH} vs the ORR activity on n(111)–(111) Pt surfaces. Pt(110) differs significantly from the linear trend of the other surfaces. Reconstructed Pt(110) shows an anomalously low ORR activity for the n(111)–(111) series. The reaction mechanism of the ORR on Pt(110) may be different from that of other n(111)–(111) surfaces. Following sentences were added in the **Discussion**. IR spectra on Pt(100) were added in **Fig. S6**.

“While the ORR activity on Pt(111) is dramatically enhanced by the presence of THA⁺, there is no significant enhancement of the ORR on Pt(100), Pt(110) and Pt(331) by THA⁺. In acidic solution above 0.6 V, the hydration water around H⁺ is linked with OH_{ad} through hydrogen bond (Fig. 5a). H⁺–H₂O–OH_{ad} formation stabilizes the OH_{ad} layer and inhibits the ORR. According to studies on the Pt(111) surface under the ultra–high vacuum condition, the coadsorption of OH_{ad} and H₂O forms a well–defined 3 × 3 honeycomb structure because the symmetry and OH•••O distance in the

coadsorbed layer fit well with the Pt(111) lattice^{14,48}. The approach of the hydration shell around THA⁺ to the Pt surface disrupts the stabilization effect between the hydration water and the OH_{ad} layer (Fig. 5b). The coverage of OH_{ad} is decreased by this destabilization effect, which induces the disruption of stable hydrogen bonding network in the coadsorbed layer. These multiple effects are enhanced by the hydrophobicity of cation and promote the access of oxygen molecules to the Pt(111) surface. The IR spectra on Pt(100) indicate that the addition of THA⁺ results in a slight reduction of the band intensity and band broadening of δ_{PtOH} as shown in Fig. S6. The charge densities for surface oxidation also decrease on Pt(110) and Pt(331) above 0.7 V after the addition of THA⁺. For Pt(100), Pt(110), and Pt(331), although OH adsorption is inhibited by the approach of the THA⁺ hydration shell, the ORR enhancement effect by the disruption of OH_{ad} + H₂O hydrogen bonding network is weak due to the narrow (111) terrace and the lattice mismatch of the OH_{ad} + H₂O honeycomb layer with the substrate. The Pt(331) surface having a (111) terrace shows high ORR activity in HClO₄ without TAA⁸. DFT calculations suggest that the OH_{ad} on stepped Pt surfaces is destabilized by deforming the hydrogen bonds of interfacial water surrounding the adsorption sites and the destabilization of oxygen species cause the enhancement of the ORR on the (111) terrace⁴⁹.”

Comment 2

A minor suggestion is to change X-ray diffraction to X-ray scattering as the specular CTR is considered as a X-ray scattering technique.

Reply

We revised from “X-ray diffraction” to “X-ray scattering” or ”X-ray CTR scattering”.

Reviewer 3

Comment 1

In Figure 1c, the authors found THA^+ shows highest activity for the ORR. However, the detailed mechanism is not written in the manuscript. The effects of the alkyl chains of TAA^+ should be discussed.

Reply

The coordination energies of hydration water with TAA cations depends on the alkyl chain length of TAA^+ . As shown in Fig. 4, we found the strong relationship between the hydration enthalpy estimated from Ref.25 and the ORR activity except TMA^+ . Furthermore, this trend corresponds to the relationship between the ORR activity and the hydration energy of alkaline metal cations in alkaline solutions (Ref. 19). Thus, the hydrophobicity strength depending on the alkyl chain length is one of the enhancement factors for the ORR activity. We also mentioned the reason the deviation of TMA^+ . Following sentences were added in the **Discussion** and the relationship between the hydration energy and the ORR activity was added in **Figure 4**.

“Since the enthalpy of hydration water with TAA^+ is reduced by an increase of the alkyl chain length²⁵, the enhancement of hydrophobicity with longer alkyl chains strongly perturbs the interfacial water, changing the equilibrium potentials for the adsorption of hydrogen and Pt oxidation on Pt(111).”

“The enhancement of the ORR activity by the addition of TAA^+ is related with the hydrophobicity of TAA^+ . Figure 4 shows the correlation between hydration enthalpy and the ORR activity. A linear correlation is found except TMA^+ and the slope for TAA^+ s is similar to that for alkali metal cations, as reported by Markovic et al¹⁹.”

“The ORR activity in TMA^+ is comparable to that in HClO_4 without TAA^+ , indicating that TMA^+ does not contribute the enhancement of the ORR. Since the hydration shell size of TMA^+ is smaller than those of the other TAA^+ s, the complete hydrogen–bonding network may not be constructed in the shell.”

Comment 2

The effects of the surface structure (Fig. 1d) of Pt are not discussed. The mechanism why THA^+ is effective only for Pt(111) should be discussed.

Reply

It is known that the OH_{ad} species is deactivation species for the ORR and the OH_{ad} is stabilized by the interaction with hydrophilic cation. However, on Pt(111), a stable hydrogen bonding network of $\text{OH}_{\text{ad}} + \text{H}_2\text{O}$ coadsorbed layer also inhibits the ORR. Thus, the ORR activity depends on the stability of hydrogen bonding network as well as the OH_{ad} coverage. On the (111) terrace, the former factor is

dominant and the ORR is significantly enhanced on Pt(111) for the following reasons. Study on n(111)–(111) Pt surfaces shows that the (111) terrace edge enhances the ORR activity. DFT calculations predict that the (111) terrace edge changes the structure of hydrogen bonding network on n(111)–(111) Pt surfaces. The structural change of the coadsorbed layer hinders the formation of Pt oxides that are known to deactivate the ORR, resulting in a higher ORR activity than Pt(111) electrodes without a terrace edge. The approach of the hydrophobic cation destabilizes the OH_{ad} layer, which causes the deformation of the stable hydrogen bonding network on Pt(111). Similar activation effect with n(111)–(111) is expected to occur on Pt(111). IR spectra of adsorbed water also suggest the change in hydrogen bonding structure.

The (111) terrace in (110) and (331) is not wide enough to form the hydrogen bonded honeycomb structure, and the 4-fold symmetry on Pt(100) does not fit in well with the honeycomb structure. We previously reported the band intensity of the δ_{PtOH} vs the ORR activity on n(111)–(111) Pt surfaces. Pt(110) differs significantly from the linear trend of the other surfaces. Reconstructed Pt(110) shows an anomalously low ORR activity for the n(111)–(111) series. The reaction mechanism of the ORR on Pt(110) may be different from that of other n(111)–(111) surfaces. Following sentences were added in the **Discussion**. IR spectra on Pt(100) was added in **Fig. S6**.

“While the ORR activity on Pt(111) is dramatically enhanced by the presence of THA⁺, there is no significant enhancement of the ORR on Pt(100), Pt(110) and Pt(331) by THA⁺. In acidic solution above 0.6 V, the hydration water around H⁺ is linked with OH_{ad} through hydrogen bond (Fig. 5a). H⁺–H₂O–OH_{ad} formation stabilizes the OH_{ad} layer and inhibits the ORR. According to studies on the Pt(111) surface under the ultra-high vacuum condition, the coadsorption of OH_{ad} and H₂O forms a well-defined 3 × 3 honeycomb structure because the symmetry and OH•••O distance in the coadsorbed layer fit well with the Pt(111) lattice^{14,48}. The approach of the hydration shell around THA⁺ to the Pt surface disrupts the stabilization effect between the hydration water and the OH_{ad} layer (Fig. 5b). The coverage of OH_{ad} is decreased by this destabilization effect, which induces the disruption of stable hydrogen bonding network in the coadsorbed layer. These multiple effects are enhanced by the hydrophobicity of cation and promote the access of oxygen molecules to the Pt(111) surface. The IR spectra on Pt(100) indicate that the addition of THA⁺ results in a slight reduction of the band intensity and band broadening of δ_{PtOH} as shown in Fig. S6. The charge densities for surface oxidation also decrease on Pt(110) and Pt(331) above 0.7 V after the addition of THA⁺. For Pt(100), Pt(110), and Pt(331), although OH adsorption is inhibited by the approach of the THA⁺ hydration shell, the ORR enhancement effect by the disruption of OH_{ad} + H₂O hydrogen bonding network is weak due to the narrow (111) terrace and the lattice mismatch of the OH_{ad} + H₂O honeycomb layer with the substrate. The Pt(331) surface having a (111) terrace shows high ORR activity in HClO₄ without TAA⁺

⁸. DFT calculations suggest that the OH_{ad} on stepped Pt surfaces is destabilized by deforming the hydrogen bonds of interfacial water surrounding the adsorption sites and the destabilization of oxygen species cause the enhancement of the ORR on the (111) terrace⁴⁹.”

Comment 3

In Fig. 2, the authors are describing about the intensity changes of the bands at 1650 and 1620 cm⁻¹, and proposed OH(ad) are formed above 0.5 V. Hydrogen-bonding networks at the electric-double layers are very sensitive to the orientation of adsorbed water molecules. The orientation of water depends on the applied potentials; therefore, the effects of the electric fields should be also discussed.

Reply

We agree with your comment. The reorientation of water by the electric fields also change the band intensity. The bands at 1650 and 1620 cm⁻¹ include the coverage change and the reorientation. However, it is difficult to quantitatively isolate each component of the band intensity. Following sentences were added in the **IR measurement of adsorbed species**.

“The increase in the intensity of δ_{HOH} above 0.5 V also involves the orientation change of water induced by the electrode potential because the water dipole responds sensitively to the electric field in the EDL.”

Comment 4

In page 7, it seems that the concentration of THA⁺ on Pt electrodes does not change from 0.3 to 1.0 V. THA⁺ is a positively charged molecule, why did not change ?

Reply

Recent X-ray scattering studies reveal the detailed electrical double layer including the OHP cation. On Ag(100) in CsBr solution (pzc ~ -0.9 V vs Ag/AgCl), the Cs cation forms the OHP layer above c(2x2)-Br layer between -0.6 V and -0.03 V vs Ag/AgCl. Similar OHP cation layer at positive potential was reported on Ag(111) in KOH, Cu(100) in H₂SO₄+KCl, and Pt(111) in alkaline solutions. These studies indicate that the OHP cation is located at the potential more positive than the pzc because of the non-covalent interactions between adsorbed anion (partially negatively charged) and the OHP cation. The amount of the OHP cation is balanced with that of adsorbed anion and the surface charge in the electrical double layer. Following sentences were added in the **Discussion**.

“The OHP cations interact with the charged species in the EDL as well as the substrate. Therefore, cations electrostatically interacting with adsorbed anions can approach the surface at the potentials more positive than the potential of zero charge (pzc)^{17,20,46,47}. The coverage of the OHP cations is balanced with that of the adsorbed anions and the surface charge depending on the electrode potential.

On the surface at a constant coverage of adsorbed counter anions, the coverage of the OHP cation decreases with increasing potential¹⁷. As described above, the IR spectra indicate that the coverage of THA^+ is approximately constant in the potential region examined in this study. However, the coverage of the negatively charged species, i.e. OH_{ad} and O_{ad} , increases with increasing potential, which compensates for the decrease of the THA^+ .”

REVIEWERS' COMMENTS:

Reviewer #2 (Remarks to the Author):

I believe that the authors convincingly explained why the cation effect strongly depends on lattice facet even though no specific adsorption is involved. The answer is based on that hydrogen bonding network is strongest on (111) facet due to symmetry match. This is consistent with other studies that showed water co-adsorption is facet sensitive and can have important effect on ORR. For example, a recent JACS paper [F. Lu, et al., 2017, 139, 7310–7317] showed the effect of adsorbed water for 2e or 4e ORR on Au facets.

A minor question is on the two sentences started on page 10 of the revised manuscript "As described above, the IR spectra indicate that the coverage of THA⁺ is approximately constant in the potential region examined in this study. However, the coverage of the negatively charged species, i.e. OH^{ad} and O^{ad}, increases with increasing potential, which compensates for the decrease of the THA⁺." Is the coverage of THA⁺ coverage constant or decreasing with increasing potential? The authors may intend to say that increased anion adsorption is balanced by positive surface charge so that THA⁺ coverage remains constant. Please consider revising for clarity before publishing.

Reviewer #3 (Remarks to the Author):

The authors precisely replied to my comments and questions, thus I think this paper is now publishable. I recommend publication of this paper to Nature Communication without further revision.

To reviewers

Thank you for the valuable comments for the improvement of our manuscript. The manuscript has revised according to the comment. The revised part is painted in yellow in the manuscript.

Reviewer 2**Comment**

A minor question is on the two sentences started on page 10 of the revised manuscript “As described above, the IR spectra indicate that the coverage of THA^+ is approximately constant in the potential region examined in this study. However, the coverage of the negatively charged species, i.e. OH_{ad} and O_{ad} , increases with increasing potential, which compensates for the decrease of the THA^+ .” Is the coverage of THA^+ coverage constant or decreasing with increasing potential? The authors may intend to say that increased anion adsorption is balanced by positive surface charge so that THA^+ coverage remains constant. Please consider revising for clarity before publishing.

Reply

As you pointed out, THA coverage does not depend on the electrode potential by the balancing of surface charge and adsorbed anion. We clearly described about THA^+ coverage as follows.

“The increase in the coverage of the negatively charged species, i.e. OH_{ad} and O_{ad} , is balanced by positive surface charge so that the coverage of the THA^+ remains constant.”